# Information Shaping for Enhanced Goal Recognition of Partially-Informed Agents

**Sarah Keren,   Haifeng Xu,   Kofi Kwapong,   David Parkes,   Barbara Grosz**

School of Engineering and Applied Sciences
Harvard University
skeren@seas.harvard.edu, hxu@seas.harvard.edu, kwapongk@college.harvard.edu, parkes@eecs.harvard.edu, grosz@eecs.harvard.edu

## Abstract

We extend goal recognition design by considering a two-agent setting in which one agent, the *actor*, seeks to achieve a goal but has only partial information about its environment. The second agent, the *recognizer*, has perfect information and aims to recognize the actor's goal from its behavior as quickly as possible. As a one-time offline intervention the recognizer can selectively reveal information to the actor. The problem of selecting which information to reveal, which we call *information shaping*, is challenging because the space of information shaping options may be extremely large, and because more information revelation need not make an agent's goal easier to recognize. We formally define this problem, and suggest a pruning approach for efficiently searching the space of information shaping options. We demonstrate the ability to facilitate recognition via information shaping and the efficiency of the suggested method on a set of standard benchmarks.

## Introduction

Goal recognition is the task of detecting the goal of agents by observing their behavior (Cohen, Perrault, and Allen 1981; Kautz and Allen 1986; Ramirez and Geffner 2010; Carberry 2001; Sukthankar et al. 2014). We consider a two-agent goal recognition setting, where the first agent, the *actor*, has partial information about a deterministic environment and seeks to achieve a goal. The second agent, the *recognizer*, has perfect information, and tries to deduce the actor's goal as early as possible, by analyzing the actor's behavior.

As a one time offline intervention, and with the objective of facilitating the recognition task, the recognizer can apply a limited number of *information shaping* modifications, implemented as changes to the actor's sensor model. Such modifications can help to differentiate the actor's behavior for different goals, potentially making it easier to interpret.

The ability to quickly understand what an agent is trying to achieve, without expecting it to explicitly communicate its objectives, is important in many applications. For example, in an assistive cognition setting (Kautz et al. 2003), it may be critical to know as early as possible when a visually impaired user is approaching a hot oven, giving the system time to react to the dangerous situation ( e.g., by calling for help, reducing the heat, etc.). In security applications it may be important to early detect users aiming at a specific destination (Boddy et al. 2005), giving the system enough time to send human agents to further investigate potential threats. Early detection is also important in human-robot collaborative settings (Levine and Williams 2014), where a robot aims to recognize what component a human user is trying to assemble, so it can gather the tools needed for the task in a timely fashion. Common to all these settings, is that agents have *incomplete information* about their environment. This affects their behavior and is key to the ability to interpret it. In addition, these settings can be controlled and modified in various ways. Specifically, it may be possible to modify an agent's behavior by manipulating its knowledge and its need to act in order to acquire new information. Such manipulations may induce behaviors that can be quickly associated to a specific goal. To demonstrate, in an assisted cognition setting, an auditory signal can inform users about a hot oven. Early notification potentially causes users aiming at a different goal (e.g., the cupboard) to move away from the oven, supporting early recognition of dangerous situations.

This work extends the *goal recognition design* (GRD) framework, which deals with redesigning agent settings in order to facilitate early goal detection (Keren, Gal, and Karpas 2014; Wayllace et al. 2016). Until now, GRD work has assumed that agents have perfect knowledge of their environment. In this paper, we extend the framework to support agents with incomplete knowledge. Specifically, we focus on GRD in deterministic environments, and use contingent planning (Bonet and Geffner 2011; Brafman and Shani 2012a; Muise, Belle, and McIlraith 2014; Albore, Palacios, and Geffner 2009) to represent the actor. The design objective is to minimize *worst case distinctiveness* (*wcd*) (Keren, Gal, and Karpas 2014), which represents the longest sequence of actions (or path cost) that is possible before the actor's goal is recognized. Note that in some instances the goal may remain unrecognized, and even go unattained, in which case the *wcd* is simply the number of actions (or accumulated action cost) until the end of execution.

To minimize *wcd* we use *information shaping* and require that the information conveyed to the actor is truthful and cannot mislead. Specifically, we use *sensor extensions* to improve information about the value of some environment variables. This is a challenging problem because the number of possible design options may be extremely large. Also, as we demonstrate below, more information need not make an agent's goal easier to recognize.

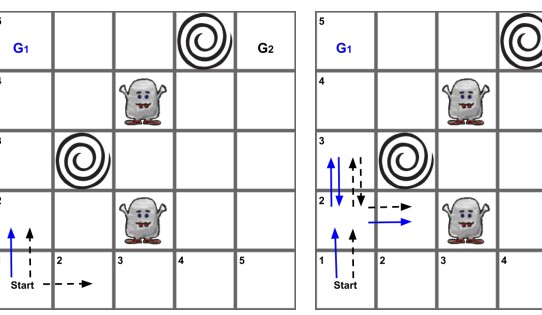
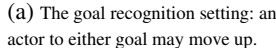
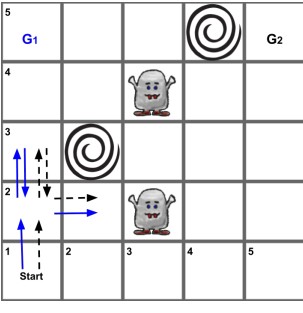
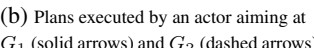
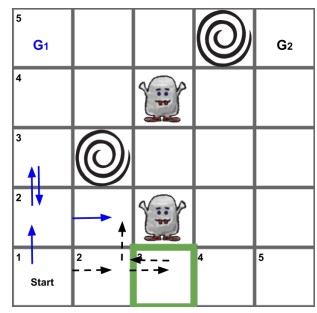
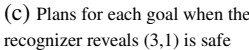
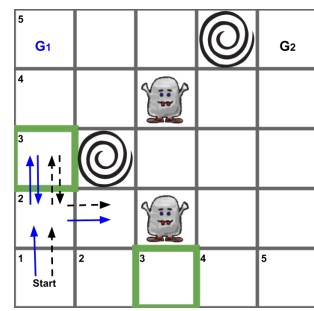

(a) The goal recognition setting: an actor to either goal may move up.

(b) Plans executed by an actor aiming at $G_1$ (solid arrows) and $G_2$ (dashed arrows).

(c) Plans for each goal when the recognizer reveals (3,1) is safe

(d) Plans for each goal when the recognizer reveals both (3,1) and (1,3) are safe.

Figure 1: An example of a GRD-APK problem

**Example 1** *As a simple example, consider Figure 1(a), depicting a variation of the Wumpus domain (Russell and Norvig 2016), where a partially informed actor has one of two goals (indicated by $G_1$ and $G_2$ in the image), and needs to achieve it without falling into pits or encountering a deadly wumpus. The actor knows its current position, but initially does not know the locations of the pits and wumpuses. When in a cell adjacent to a pit, it senses a 'breeze' and it can smell the stench of a wumpus from an adjacent cell. The recognizer has perfect information: it knows the locations of the actor, the pits (e.g., the spiral at cell $(2,3)$) and the wumpuses (e.g., cell $(3,2)$).*

*The actor starts at 'Init'. With no breeze or stench, it deduces the adjacent cells are* safe. *In this example, we will assume the actor is optimistic when planning but conservative when acting (Bonet and Geffner 2011). For planning, the actor makes the most convenient assumptions about (chooses the value of) unknown variables, plans accordingly, and revises the assumptions and re-plans if these assumptions are refuted during execution. If there are multiple cost-minimal plans (under optimism), we assume the actor selects one that requires making as few assumptions as possible (and arbitrarily otherwise). Consequently, an agent aiming at $G_1$ will start by moving up. In contrast, an uninformed agent aiming at $G_2$ is indifferent to going up or right, and may go either way. Because of this, moving up from the initial state leaves the goal unrecognized. Let us suppose (Figure 1b) that plans to both goals start by moving up two steps. After sensing a breeze at cell (1,3), not knowing which adjacent cells have a pit, the actor backtracks and moves right. After sensing a 'breeze' and 'stench', the actor deduces there is a wumpus at cell (3,2), and realizes that it will sense a stench at cell (3,1), without having the option of verifying that cell (4,1) is safe. With no more cells to explore, it halts at (2,2) leaving the goal unrecognized even after it terminates execution, setting* wcd *to* 4.

*To promote early recognition, the recognizer can share information with the actor before it starts execution, for example by revealing safe cells. However, suppose there is a budget, limiting the number of facts that can be revealed. If the recognizer chooses to reveal cell $(3,1)$ is safe (Figure 1(c)), an actor aiming at $G_2$ (originally indifferent to moving up or right) prefers moving right from the initial state. In contrast,*

*an actor aiming at $G_1$ still prefers moving up. The goal of the actor becomes clear as soon as the first step is performed and* wcd *is minimized (*wcd=0*). Note that if, in addition, the recognizer reveals that cell $(1,3)$ is safe (Figure 1(d)), the initial situation is recovered, since an actor to either goal may now choose to move up given its beliefs about minimal plans. This illustrates the need to carefully select the information to reveal in order to facilitate the recognition task.*

The contributions of this work are fourfold. First, we extend the GRD framework to support agents with partial information. We refer to our extended setting as *GRD for Agents with Partial Knowledge* (GRD-APK), and suggest information shaping modifications that can be applied to support goal recognition. Second, since our extended design setting induces a large search space of possible information shaping modifications and since previous approaches to design do not apply to our setting, we present a novel pruning method, and specify the conditions under which it is safe, so that at least one optimal solution is not pruned. Third, we implement our suggested approach, using STRIPS (Fikes and Nilsson 1972) to represent our generic and adaptable re-design process. Finally, we evaluate the algorithm on a set of standard benchmarks, and demonstrate both *wcd* reduction achievable through information shaping and the efficiency of our approach.

## Background: Planning Under Partial Observability

To support agents with partial knowledge, we follow Bonet and Geffner (2011) and consider contingent planning under partial observability, formulated as follows.

**Definition 1** *A **planning under partial observability with deterministic actions (PPO-det) problem** is a tuple $P = \langle \mathcal{F}, \mathcal{A}, I, G, \mathcal{O} \rangle$ where $\mathcal{F}$ is a set of fluent symbols, $\mathcal{A}$ is a set of actions, $I$ is a set of* clauses *over fluent-literals defining the initial situation, $G$ is a set of fluent-literals defining the goal condition, and $\mathcal{O}$ represents the agent sensor model.*

An action $a \in \mathcal{A}$ is associated with a set of preconditions $prec(a)$, which is the set of fluents that need to hold for $a$ to be applicable, and conditional effects $eff(a)$, which is a set

of pairs $(\mathcal{F}_{cond}, \mathcal{F}_{eff})$ s.t. $\mathcal{F}_{eff} \subseteq \mathcal{F}$ become true if $\mathcal{F}_{cond} \subseteq \mathcal{F}$ are true when $a$ is executed.

The sensor model $\mathcal{O}$ is a set of observations $o \in \mathcal{O}$ represented as pairs $(C, L)$ where $C$ is a set of fluents and $L$ is a positive fluent, indicating that the value of $L$ is observable when $C$ is true. Each observation $o = (C, L)$ can be conceived as a sensor on the value of $L$ that is activated when $C$ is true.

A state $s$ is a truth valuation over the fluents $\mathcal{F}$ ('true' or 'false'). For an agent, the value of a fluent may be known or unknown. A fluent is *hidden* if its true value is unknown. A *belief state* $b$ is a non-empty collection of states the agent deems possible at some point. A formula $\mathbb{F}$ *holds* in $b$ if it holds for every state $s \in b$. An action $a$ is *applicable* in $b$ if the preconditions of $a$ hold in $b$, and the *successor* belief state $b'$ is the set of states that results from applying the action $a$ to each state $s$ in $b$. When an observation $o = (C, L)$ is activated, the successor belief is the *maximal* set of states in $b$ that agree on $L$. The initial belief is the set of states that satisfy $I$, and the goal belief are those that satisfy $G$. A formula is *invariant* if it is true in each possible initial state, and remains true in any state that can be reached from the initial state. A *history* is a sequence of actions and beliefs $h = b_0, a_0, b_1, a_1, \ldots, b_n, a_n, b_{n+1}$. It is *complete* if the performing agent reaches a goal belief state.

A solution to a PPO-det problem $P$ is a *policy* $\pi$, which is a partial function from beliefs to actions. A policy is *deterministic* if any belief $b$ is mapped to at most one action. Otherwise it is *non-deterministic*. A history $h$ **satisfies** $\pi$, if $\forall i \ 0 \le i \le n, a_i \in \pi(b_i)$. There are three types of policies: *weak*, when there is at least one complete history that satisfies the policy, *strong*, where a goal belief is guaranteed to be achieved within a fixed number of steps, and *strong cyclic*, where a goal belief is guaranteed to be achieved, but with no upper bound on the cost (length) of the solution. Our framework, suggested next, supports all three policy types.

# Goal Recognition Design for Agents with Partial Knowledge (GRD-APK)

The *goal recognition design for agents with partial knowledge problem* (GRD-APK) consists of an initial goal recognition setting, a measure by which a setting is evaluated, and a design model, which specifies the information shaping modifications that can be applied. We first define each component separately.

## Goal Recognition

A goal recognition setting can be defined in various ways (Sukthankar et al. 2014), but typically includes a description of the underlying environment, the way agents behave in it to achieve their goal, and the observations collected by the goal recognizing agent. Accordingly, our goal recognition model supports two agents; a partially informed contingent planning *actor* (Definition 1) with a goal, that executes history $h$ until reaching a goal belief or halting when no action is applicable. The second agent is a perfectly informed *recognizer*, that analyzes the actor's state transitions in order to recognize the actor's goal.

**Definition 2** *A **goal recognition for agents with partial knowledge problem** (GR-APK) is a tuple $R = \langle E, \mathcal{G}, \mathcal{O}^{ac}, \{\Pi(G)\}_{G \in \mathcal{G}} \rangle$ where:*
- $E = \langle \mathcal{F}, \mathcal{A}, I \rangle$ *is the environment, which consists of the fluents $\mathcal{F}$, actions $\mathcal{A}$ and initial state $I$ as defined in Definition 1 (a cost $\mathcal{C}(a)$ for each action $a \in \mathcal{A}$ may also be specified),*
- $\mathcal{G}$ *is a set of possible goals $G$, s.t. $|\mathcal{G}| \ge 2$ and $G \subseteq \mathcal{F}$,*
- $\mathcal{O}^{ac}$ *is the actor's sensor model (Definition 1), and*
- $\{\Pi(G)\}_{G \in \mathcal{G}}$ *are the set of policies $\Pi(G)$ an agent aiming at goal $G \in \mathcal{G}$ may follow.*

The cost of history $h$, denoted $\mathcal{C}_a(h) = \Sigma_i \mathcal{C}(a_i)$, is the accumulated cost of the performed actions (equal to path length when action cost is uniform). In executing $h$, the actor follows a possibly non-deterministic policy $\pi$ from the set $\Pi(G)$ of possible policies to its goal.

The set $\Pi(G)$ of policies to each goal is typically implicitly defined via the solver used by the actor to decide how to act in each belief state. In Example 1 we described an example of such a solver, which we will formally define in the next section. The GRD-APK framework is well defined for any solver that provides a mapping $\mathcal{B} \to 2^{\mathcal{A}}$, specifying the set of possible actions an agent may execute at each reachable belief state $b \in \mathcal{B}$ (e.g., (Bonet and Geffner 2011; Muise, Belle, and McIlraith 2014).

In our setting, the actor and recognizer both know the environment $E$ and the set $\mathcal{G}$ of possible goals. While the partially informed actor needs to collect information about the environment via its sensor model $\mathcal{O}^{ac}$ in order to achieve its premeditated goal, the recognizer knows the true state of the world and the actor's solver and sensor, but does not know the actor's goal. The recognizer observes the actor's transitions between belief states and analyzes them in order to recognize the actor's goal.[1]

## Evaluating a GR-APK model

The *worst case distinctiveness* (*wcd*) measure represents the maximum number of actions an actor can perform (in general, maximum total cost incurred by the actor) before its goal is revealed. To define *wcd* we first define the relationship between the observations collected by the recognizer when an actor follows history $h$, which in our case correspond to the actor's transitions between belief states, and a goal. As mentioned above, we say that a history *satisfies* a policy, if it is a possible execution of the policy. In addition, a history *satisfies* a goal, if satisfies a possible policy to the goal.

**Definition 3** *Given a GR-APK model $R$, history $h$ **satisfies** policy $\pi$ in $R$, if $\forall i \ 0 \le i \le n, a_i \in \pi(b_i)$. In addition, $h$ **satisfies** goal $G \in \mathcal{G}$ in $R$ if $\exists \pi \in \Pi(G)$ s.t. $h$ satisfies $\pi$.*

---

[1]Since we are analyzing the goal recognition setting, and need to account for all possible observations of agent behavior, we do not specify a particular history to be analyzed, which is a typical component in goal recognition models (e.g., (Ramirez and Geffner 2010; Pereira, Oren, and Meneguzzi 2017). Instead, in facilitating goal recognition via design, our model characterizes the different actor behaviors in the system, and the way they are perceived by the recognizer.

Let $\mathcal{G}^{rec}(h)$ represent the set of goals that history $h$ satisfies, i.e., the set of goals the recognizer deems as possible actor goals. We define a history as *non-distinctive* if it satisfies more than one goal.

**Definition 4** *Given a GR-APK model $R$, a history $h$ is **non distinctive** in $R$, if exists $G, G' \in \mathcal{G}$ s.t. $G \neq G'$, and $h$ satisfies $G$ and $G'$. Otherwise, it is distinctive.*

We denote the set of non-distinctive histories of a GR-APK model $R$ by $H^{nd}(R)$.

**Definition 5** *The **worst case distinctiveness** of a model $R$, denoted by $\mathrm{wcd}(R)$ is:*

$$\mathrm{wcd}(R) = \begin{cases} \max\limits_{h \in H^{nd}(R)} \mathcal{C}_a(h) & H^{nd}(R) \neq \emptyset \\ 0 & otherwise \end{cases}$$

That is, *wcd* is the maximum cost history for which the goal is not determined, or zero if there is no such history. Recall that in some instances the goal may remain unrecognized, and even go unattained, in which case the *wcd* is simply the number of actions (or accumulated action cost) until the end of execution. Also recall that a policy may be strong cyclic, potentially containing infinite loops. A policy with such a cycle is considered to have a history with infinite cost. In particular, since such a history may be non-distinctive, this means *wcd* in this setting may be infinite.

### Information Shaping

Our interest here is in modulating the behavior of the actor through information shaping. By changing the actor's knowledge, we can potentially change its behavior and the way by which it acquires the information needed to achieve it's goal. We restrict the information shaping interventions to be truthful so that they cannot convey false information. In the context of contingent, partially-informed planning agents, this requirement is naturally implemented by requiring that we may only improve the actor's sensor model, i.e., improving its ability to access the value of some environment feature. We define *sensor extension* modifications, which add a single observation to a sensor model, using $\mathcal{O}$ to denote the set of all sensor models.

**Definition 6** *A modification $\delta : \mathcal{O} \to \mathcal{O}$ is a **sensor extension** if $\delta(\mathcal{O}) = \mathcal{O} \cup \{o\}$, for all $\mathcal{O} \in \mathcal{O}$, and for some $o = (C, L)$.*

Sensor extensions correspond to adding new sensors to the environment, or, as a special case, communicating to the actor the value of a feature (setting $C = \emptyset$).

To demonstrate, in Example 1 the recognizer can allow the actor to sense a stench in cell $(1, 2)$, two (rather than one) cells away from the wumpus in cell (3,2). This extension is implemented by adding the observation $o = (C = AgentAtCell(1, 2), L = BreezeInCell(2, 2))$ to the actor's sensor model. This could be realized through a visual indication or sign, similar to the auditory signal indicating the oven is hot in the assisted cognition example. The recognizer could also directly communicate with the

actor and inform it about the location of a wumpus, or reveal a location without a wumpus. (e.g., $(C = True, L = WumpusAtCell(4, 4))$).

We are now ready to define a GRD-APK problem.

**Definition 7** *A **goal recognition design for agents with partial knowledge problem** (GRD-APK) is defined as a tuple $T = \langle R_0, \mathbf{\Delta}, \beta \rangle$ where:*
- $R_0$ *is the initial goal recognition model,*
- $\mathbf{\Delta}$ *are the possible sensor extensions, and*
- $\beta$ *is a budget on the number of allowed extensions.*

We want to find a set $\Delta \subseteq \mathbf{\Delta}$ of up to $\beta$ sensor extensions to apply to $R_0$ offline to minimize the *wcd*. This objective is formally defined below, where $wcd^{min}(T)$ is the minimum *wcd* achievable in a GRD-APK model $T$, and $R^\Delta$ is the goal recognition model that results from applying set $\Delta$ to $R$.

$$wcd^{min}(T) = \min_{\Delta \subseteq \mathbf{\Delta}} \quad wcd(R_0^\Delta)$$
$$s.t. |\Delta| \leq \beta \tag{1}$$

Any solution to Equation 1 is *optimal*, i.e., it achieves the minimal *wcd* possible. It is *strongly optimal* if it has minimum size among all optimal solutions, i.e., it includes the minimal number of extensions needed to minimize *wcd*.

### The *k-planner* and $K_{prudent}(P)$ Translation

A variety of solvers have been developed to solve a PPO-det problem (e.g., (Bonet and Geffner 2011; Muise, Belle, and McIlraith 2014; Brafman and Shani 2012b)), all of which can be used to represent the actor (and its set of possible policies) described in Definition 2. Specifically, Bonet and Geffner (2011) suggest the *k-planner* that follows the *planning under optimism* approach; the actor plans while making the most convenient assumptions about the values of (i.e., assigns a value to) hidden variables, executes the plan that is obtained from the resulting classical planning problem, and revises the assumptions and re-plans, if during the execution, an observation refutes the assumptions made.

To transform the PPO-det problem into a classical planning problem, the k-planner uses the $K(P)$ translation. At the core of the translation is the substitution of each literal $L$ in the original problem with a pair of fluents $KL$ and $K\neg L$, representing whether $L$ is known to be true or false, respectively (Albore, Palacios, and Geffner 2009). Each original action $a \in \mathcal{A}$ is transformed into an equivalent action that replaces the use of every literal $L$ ($\neg L$), with its corresponding fluent $KL$ ($K\neg L$). Each observation $(C, L)$ is translated into two deterministic sensing actions, one for each possible value of $L$. These sensing actions allow the solver to compute a plan while choosing preferred values of (making assumptions about) the unknown variables. For example, the actor can assume that a cell on its planned path has no pit (e.g., $K\neg PitAt(4, 1) = True$). Each invariant clause is translated into a set of actions, which we call *ramification actions*. These actions can be used to set the truth value of some variable, as new sensing information is collected from the environment. For example, a ramification action can be activated to deduce that a cell is safe when no breeze or stench is sensed in an adjacent cell.

The action set in the transformed problem is therefore $\mathcal{A}' = \mathcal{A}'_{exe} \cup \mathcal{A}'_{sen} \cup \mathcal{A}'_{ram}$, where $\mathcal{A}'_{exe}$ represents the transformed original set of actions, $\mathcal{A}'_{sen}$ are the sensing actions and $\mathcal{A}'_{ram}$ are the ramification actions. This representation captures the underlying planning problem at the knowledge level, accounting for the exploratory behavior of a partially informed agent.

Bonet and Geffner (2011) show that this linear translation of a PPO-det problem into a classical planning problem is sound and complete for *simple* PPO-det models with a connected state space. A PPO-det model is simple if the non-unary clauses in $I$ are all invariant, and no hidden fluent appears in the body of a conditional effect. In connected state spaces every state is reachable from any other. In simple problems there is no information loss and the model is *monotonic*, i.e., for every fluent $f \in \mathcal{F}$, if $f$ is known in a belief state $b$ and $b'$ is a belief reachable from $b$, then $f$ is known in $b'$. As a consequence, for every policy $\pi$ and history $h$ of length $n$ it follows that the number of states in beliefs $b_i$ is a monotonically decreasing function, i.e., $|b_i| \geq |b_{i+1}|$ for every $0 \leq i < n$.

A key issue to note about the $K(P)$ compilation is that all its actions, including sensing and ramification actions, have equal cost. This means that a cost-minimizing solution to the resulting classical planning problem may be one that favors increasing the cost to goal over the use of multiple ramification actions. As described in Example 1, we want a solver that can make optimistic assumptions, but chooses a minimal cost plan that requires making as few assumptions as possible. In addition, ramifications are not to be considered when calculating the cost to goal. We therefore suggest the $K_{prudent}(P)$ translation, which extends the uniform cost $K(P)$ translation by associating a cost function to each action in $\mathcal{A}'$. Specifically, every transformed action $a \in \mathcal{A}'_{exe}$ is assigned a cost of 1, every sensing action (assumption) $a \in \mathcal{A}'_{sen}$ is assigned a small cost of $\epsilon$, and every ramification action $a \in \mathcal{A}'_{ram}$ has 0 cost. When $\epsilon$ is small enough such that the accumulated cost of assumptions of any generated plan is guaranteed to be smaller than minimal diversion from an optimal plan, the cost-minimal plan achieved using this formulation complies with our requirements.

## Methods for Information Shaping

In our search for an optimal design solution, we consider a sensor extension as *useful* with regards to a goal recognition model if it reduces *wcd*. Given a goal recognition model $R$ and a sensor extension $\delta$, we let $R^\delta$ denote the model that results from applying $\delta$ to the actor's sensor model $\mathcal{O}$, and define useful sensor extensions as follows.

**Definition 8** *A modification $\delta$ is* **useful** *with regards to goal recognition model $R$ if* $wcd(R^\delta) < wcd(R)$.

The challenge in information shaping comes from two sources. First, the number of possible information shaping options may be large, and evaluating the effect of each change may be costly, making it important to develop efficient search techniques. Second, the problem is non-monotonic, in that sensor extensions are not always useful,

and providing more information may actually make recognition more difficult by increasing *wcd* (Example 1).

To address these challenges, we follow Keren, Gal, and Karpas (2018) and formulate the design process as a search in the space of modification sets $\Delta \subseteq \boldsymbol{\Delta}$. With a slight abuse of notation, we let $R^\Delta$ denote the model that results from applying the set $\Delta$ of sensor extensions to the actor's sensor model. The root node is the initial goal recognition model $R_0$ (and empty modification set), and the operators (edges) are the sensor extensions $\delta \in \boldsymbol{\Delta}$ that transition between models. Each node (modifications set $\Delta$) is evaluated by $wcd(R_0^\Delta)$, the *wcd* value of its corresponding model.

To calculate the *wcd* value of a model we need to find the maximal non-distinctive history. Recall that we assume the actor's solver is known to the recognizer, who can observe the actor's transition between states. We can therefore find the *wcd* value of a GR-APK model by first using the actor's solver to compute the policies to each of the goals. Then, starting at the initial state, we iteratively explore the non-distinctive policy prefixes, until its most distant boundary is found, and return its length (cost).

## Design with CG-Pruning

The baseline approach for searching in modification space is *breadth first search* (BFS), using *wcd* to evaluate each node. Under the budget constraints, BFS explores modification sets of increasing size, using a closed-list to avoid the computation of pre-computed sets. The search halts if a model with $wcd = 0$ is found or if there are no more nodes to explore, and returns the shortest path (smallest modification set) to a node that achieves minimal *wcd*. This iterative approach is guaranteed to find a strongly optimal solution, i.e., a minimal set of modifications that minimizes *wcd*. However, it does not scale to larger problems.

To increase efficiency, pruning can be applied to reduce the size of the search space. Specifically, pruning is *safe* if at least one optimal solution remains unpruned (Wehrle and Helmert 2014). Keren, Gal, and Karpas (2018) offer a pruning technique for GRD settings where the actor is fully informed and guarantee it is safe if modifications cannot increase *wcd*. Since this condition does not hold in our setting, where sensor extensions can both increase and reduce *wcd*, we suggest a new pruning approach that eliminates useless modifications, and specify conditions under which it is safe.

The high level idea of our pruning technique is to transform the partially observable planning problem for each goal into its corresponding fully observable planning problem, and use off-the-shelf tools developed for fully observable planning in order to automatically detect information shaping modifications that are guaranteed not to have an effect on the actor's behavior.

Specifically, given a goal recognition model $R$, for every goal in $\mathcal{G}$, we use the $K(P)$ transformation (or its variant $K_{prudent}(P)$ introduced above) to transform the partially observable planning problem into a fully observable problem. We then construct the *causal graph* (Williams and Nayak 1997; Helmert 2006) of each transformed problem. According to Helmert (2006), the causal graph of a planning problem is a directed graph $(V, E)$ where the nodes

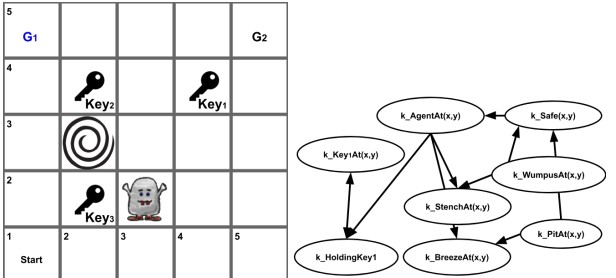

Figure 2: The Wumpus domain with keys

$V$ represent the state variables and the edges $E$ represent dependencies between variables, such that the graph contains a directed edge $(v, v')$ for $v, v' \in V$ if changes in the value of $v'$ can depend on the value of $v$. Specifically, to capture only the variables that are relevant to achieving the goal, the causal graph only contains ancestors of all variables that appear in the goal description. In our context, the variable set of the causal graph can either be the set of fluents of the transformed PPO-det problem, or the multi-valued variables extracted using *invariant synthesis*, which automatically finds sets of fluents among which exactly one is true at each state, and which can be assumed to represent the different values of a multi-valued variable. In any case, the casual graph $CG(G)$ of each goal $G \in \mathcal{G}$ captures all variables relevant for achieving the goal and the hierarchical dependencies between them. Recall that each sensor extension is characterized by an observation $o = (C, L)$ that is added to the actor's sensor model. Our pruning technique, dubbed *CG-Pruning*, prunes all sensor extensions for which the fluents corresponding to knowledge about $L$ in the transformed problem (i.e. $KL$ and $K\neg L$) do not occur in any of the casual graphs.

**Example 2** *Consider Figure 2(left), depicting a modified version of Example 1, where the actor needs to collect a key to be able to access its goal (e.g., $PickedKey_1$ is needed to reach $G_1$). There are multiple keys distributed in the grid (e.g., $Key_1At(4,4)$), each needed for accessing a particular location. The actor initially knows a set of possible key locations for each key. When in a cell with a key, it senses it and can pick it up and use it to achieve its goal. In this scenario, the recognizer, with perfect information, can notify the actor about safe locations, as before, but also about the absence or presence of a particular key in some location. Applying the $K(P)$ transformation here creates fluents $KKey_iAt(x,y)$ for each key and location, representing whether the actor knows key $i$ is at location $(x,y)$, which is a precondition to picking up the key. Figure 2(right), show a part of the causal graph for $G_1$ that only includes variables concerning the location of its relevant key. By generating the causal graph to all goals, we automatically detect and prune sensor extensions regarding variables that do not appear in any of the causal graphs (e.g., the sensor extension that reveals the location of $Key_3$).*

In the following, we show that CG-Pruning is safe for

GRD-APK settings where the actor uses the *k-planner* with an optimal planner to computes its plans. Since the actor uses the *k-planner*, it iteratively computes a policy at the initial state and every time an assumption made at a previous iteration is refuted. At each iteration, the current partially observable problem is transformed into its corresponding fully observable problem, and a new plan is computed and executed. This continues until the actor reaches a goal belief or a belief state with no applicable actions. For each model $R$ and execution iteration $i$, we let $CG_i^R(G)$ represent the causal graph at iteration $i$ and start our proof by showing that the causal graph at each iteration subsumes any causal graph of subsequent iterations.

**Lemma 1** *For any model $R$ and goal $G \in \mathcal{G}$, $CG_j^R(G)$ is a subgraph of $CG_i^R(G)$ for any $i, j$ s.t. $0 \le i < j$.*

**Proof Sketch:** The causal graph of iteration $i$ captures all variables that appear in actions that may be applied in order to achieve a goal belief from the initial belief state at iteration $i$. This graph includes all the actions (and their corresponding variables) that may be applied from the belief reached at iteration $j$. ∎

Lemma 1 guarantees that a variable that does not occur in $CG_0^R(G)$ for any goal $G \in \mathcal{G}$ will not occur in causal graphs of future iterations.

Next, we observe that when an optimal solver is used, a sensor extension that does not correspond to a variable in the initial causal graph of any goal is not useful.

**Lemma 2** *For any model $R$ and sensor extension $\delta$ that adds observation $o = (C, L)$ to $\mathcal{O}^{ac}$, if for all $G \in \mathcal{G}$, $KL$ and $K\neg L$ are not in $CG_0^R(G)$, then $\delta$ is not useful w.r.t $R$.*

**Proof Sketch:** Bonet and Geffner (2011) show that the $K(P)$ transformation is sound and complete for simple problems with a connected space, which are the only problems we consider here. Helmert (2006) shows that any optimal plan can be acquired by ignoring variables that are not in the causal graph. Therefore, by pruning sensor extensions that are related to variables not on the causal graph, we are removing from the actor's planning graph sensing actions that would anyway not appear in any optimal plan (i.e., assumptions the actor would not make). Therefore, the behavior of an actor to any goal is not affected by such sensor extensions. Moreover, as the actor progresses and re-plans, no sensing action can be added to the actor's model. Consequently, the behavior w.r.t to any goal will not change, $wcd$ will not change, and therefore $\delta$ is not useful w.r.t $R$. ∎

Finally, we are ready to show that CG-Pruning is safe.

**Theorem 1** *For any GRD-APK model $T = \langle R_0, \boldsymbol{\Delta}, \beta \rangle$, CG-Pruning is safe for an actor that uses the k-planner with an optimal planner.*

**Proof Sketch:** Lemma 2 guarantees, that under the assumptions we make, any sensor extension that adds observation

$o = (C, L)$ to $\mathcal{O}^{ac}$ and for which neither $KL$ or $K\neg L$ appear in $CG_0^R(G)$, are not useful to any model reachable from $R_0$ via design and will not be part of a strongly optimal solution. Therefore CG-Pruning is safe.

∎

## Empirical Evaluation

Our objective is to evaluate both the effect sensor extensions have on *wcd* as well as the efficiency of CG-Pruning. We start by describing our dataset and empirical setup, and then discuss our initial results.

**Dataset.** We used five domains adapted from Bonet and Geffner (2011) and Albore, Palacios, and Geffner (2009).

- WUMPUS: corresponding to the setting in Example 1.
- WUMPUS-KEY: corresponding to Example 2.
- C-BALLS (Colored-balls): the actor navigates a grid to deliver balls of different and initially unknown colors to their per-color destinations.
- TRAIL: an agent must follow a trail to reach a destination, while sensing only the reachable cells surrounding it.
- Logistics: Packages are transported to their destinations, relying on sensing to reveal the packages in a location.

The adaptation from contingent planning to GRD-APK involves specifying for each instance the set of possible goals and sensor extensions (see Table 1 for details).

To support the design process, we use STRIPS (Fikes and Nilsson 1972) to specify the available modifications (and their effect). Sensor extensions are implemented as design actions that add to the initial state fluents that represent the true value of a variable.

**Setup.** We use the *k-replanner* (Bonet and Geffner 2011) as the actor's solver, with two variations. For the first, the $K(P)$ compilation was used together with the satisfying FF classical planner (FF) (Hoffmann and Nebel 2001). The second used the $K_{prudent}(P)$ compilation together with the optimal Fast-Downward (Helmert 2006) classical planner (FD), using the lm-cut heuristic (Helmert and Domshlak 2009).

In our computation of *wcd*, we also consider the prefixes of failed executions, since they represent valid agent behavior. The design process is implemented as a breadth-first search (BFS) in the space of modification sets, tested with and without CG-Pruning.

We use 30 instances for each domain, and a design budget of $1-2$. Each execution had a time limit of 20 minutes and is capped at 1000 search steps (each corresponding to a design set), whichever was first.

To parse the design file, we adopt the parser of *pyperplan* (Alkhazraji et al. 2016), which provide for each modification set (representing a GRD-APK model and a node in our search) the set of successors (applicable modifications) and the model that results from applying each modification.

**Results.** Tables 2 and 3 summarize the results for both approaches (No Pruning vs. CG-Pruning) for the FD and FF solvers, respectively. For each domain and design budget ($b = 1$ and $b = 2$), the tables shows 'sol' as the fraction of instances completed within the time and resource bounds. For instances completed by both approaches '$\Delta$-*wcd*' is the

|  | Possible Goals | Sensor Extensions |
|---|---|---|
| **WUMPUS** | gold locations | safe cells |
| **WUMPUS-KEY** | gold locations | safe cells or locations with / without keys |
| **C-BALLS** | ball distribution | locations without a ball |
| **TRAIL** | final stone locations | locations with / without stones |
| **LOGISTICS** | package destination | package locations |

Table 1: Possible goals and design options for each domain.

|  |  | No Pruning | | | | CG-Pruning | | | |
|---|---|---|---|---|---|---|---|---|---|
|  | budget | sol | $\Delta wcd$ | time | nodes | sol | $\Delta wcd$ | time | nodes |
| **WUMPUS** | b=1 | 0.1 | 0.0 (1.8) | 92.84 | 14.0 | 0.1 | 0.0 (1.8) | 76.96 | 11.0 |
|  | b=2 | 0.1 | 0.0 (1.8) | 663.71 | 106.0 | 0.1 | 0.0 (1.8) | 421.27 | 67.0 |
| **WUMPUS-KEY** | b=1 | 1.0 | 0.2 (0.57) | 16.05 | 4.1 | 1.0 | 0.2 (0.57) | 12.59 | 3.3 |
|  | b=2 | 0.71 | 0.2 (0.57) | 238.74 | 39.5 | 0.72 | 0.2 (0.57) | 200.34 | 28.8 |
| **LOGISTICS** | b=1 | 0.14 | 8.0 (11.83) | 1330.33 | 3.0 | 0.14 | 8.0 (11.83) | 1042.52 | 2.0 |
|  | b=2 | NA | NA | NA | NA | NA | NA | NA | NA |

Table 2: Results per domain for (optimal) the FD solver

average *wcd* reduction achieved via design, i.e., the *wcd* difference between the original setting and one where sensor extensions are applied (note that since CG-Pruning is safe '$\Delta$-*wcd*' is the same for both approaches). In parenthesis we show '$\Delta$-*wcd*' over all instances, including those that timed out. The average calculation time (in seconds) for each approach is indicated by 'time', and 'nodes' is the average number of nodes evaluated on all instances. 'NA' represents settings for which no instance completed. In Table 2 we excluded C-BALLS and TRAIL, since no problem completed for both domains.

Our results show that design via information shaping reduces *wcd* for all domains, with a reduction of 9.12 (about half) for C-BALLS. By excluding futile sensor extensions, for all domains CG-Pruning reduces the number of nodes explored and computation time for completed problems. For WUMPUS, WUMPUS-KEY and LOGISTICS using FF, CG-Pruning also increases the ratio of solved problems.

The results show the potential of our pruning approach. However, many instances were not completed for FD, failing in some cases to complete the solution of the initial setting. To achieve more results for the optimal case, and hopefully a stronger indication of the benefit of our approach in such settings, we intend to add additional domains to our dataset and explore different heuristics used to guide the optimal search. We also intend to enhance pruning further. Specifically, using the plan the actor intends to execute with regards to each goal, we can prune sensor extensions that correspond to as-

|  |  | No Pruning | | | | CG-Pruning | | | |
|---|---|---|---|---|---|---|---|---|---|
|  | budget | sol | $\Delta wcd$ | time | nodes | sol | $\Delta wcd$ | time | nodes |
| **WUMPUS** | b=1 | 1.0 | 0.0 (3.0) | 88.02 | 16.0 | 1.0 | 0.0 (6.0) | 59.98 | 11.0 |
|  | b=2 | 0.25 | 0.0 (3.0) | 697.31 | 137.0 | 1.0 | 0.0 (6.0) | 351.37 | 67.0 |
| **WUMPUS-KEY** | b=1 | 1.0 | 4.33 (4.33) | 16.71 | 13.55 | 1.0 | 4.33 (4.33) | 13.35 | 10.56 |
|  | b=2 | 0.8 | 3.95 (3.95) | 85.73 | 54.56 | 1.0 | 3.95 (3.95) | 75.55 | 42.55 |
| **C-BALLS** | b=1 | 0.8 | 9.12 (9.2) | 36.61 | 37.03 | 0.8 | 9.12 (9.2) | 38.75 | 37.03 |
|  | b=2 | 0.8 | 11.5 (10.83) | 30.19 | 22.01 | 0.8 | 11.5 (10.83) | 30.19 | 22.01 |
| **TRAIL** | b=1 | 1.0 | 0.0 (0.0) | 14.71 | 28.0 | 1.0 | 0.0 (0.0) | 12.97 | 26.5 |
|  | b=2 | 1.0 | 0.0 (0.0) | 195.39 | 407.0 | 1.0 | 0.0 (0.0) | 173.21 | 365.5 |
| **LOGISTICS** | b=1 | 0.42 | 3.01 (4.14) | 22.90 | 61.5 | 1.0 | 3.01 (4.14) | 19.41 | 42.67 |
|  | b=2 | 0.28 | 9.05 (9.27) | 133.68 | 234.4 | 0.86 | 9.05 (9.17) | 112.89 | 175.1 |

Table 3: Results per domain for the (satisfying) FF solver

sumptions already made by the actor, and show that they will not reduce the *wcd*.

## Related Work

Goal Recognition Design (GRD), a special case of *environment design* (Zhang, Chen, and Parkes 2009), was first introduced by Keren et al. (2014) to account for optimal fully observable agents in deterministic domains. This work was later extended to a variety of GRD settings, including accounts for sub-optimal actors (Keren, Gal, and Karpas 2015), stochastic environments (Wayllace et al. 2016), adversarial actors that try to conceal their goal (Ang et al. 2017), and a partially informed recognizer (Keren, Gal, and Karpas 2016a; 2016b; 2018). In the latter case, sensor refinement is applied to enhance the recognizer's sensor model.

Common to all previous GRD work is the assumption that actors have perfect observability of their environment. Our work is the first to generalize GRD to account for a partially informed actor and to suggest new information shaping modifications, implemented as sensor extensions applied to the actor's sensor model, as a way to reduce *wcd*.

Efficient communication via selective information revelation is fundamental to various multi agent settings, e.g., (Xuan, Lesser, and Zilberstein 2001; Wu, Zilberstein, and Chen 2011; Unhelkar and Shah 2016; Dughmi and Xu 2016). This work is the first to use information shaping as a one time and offline intervention that is performed in order to facilitate goal recognition.

## Conclusion

We introduced GRD for a partially informed actor and a perfectly informed recognizer, who can share information about the domain with the actor. We formalized the information shaping problem as one of minimizing worst-case distinctiveness, and presented new sensor extension modifications, used to enhance recognition. We studied the use of breadth first search to search the space of applicable sensor extensions, developing a safe pruning approach to improve efficiency. To the best of our knowledge, this is the first paper to suggest using techniques developed for classical planning toward the design of algorithms for goal recognition of partially informed planning agents. Our results on a set of standard benchmarks show that *wcd* can be reduced via information shaping and demonstrate the efficiency of our approach.

There are many ways to extend this work. First, we use qualitative contingent planning models to represent the partially informed agent and its belief states. It would be interesting to extend this work to use Partially Observable Markov Descision Process (POMDP) models (Kaelbling, Littman, and Cassandra 1998) to represent the actor. Another interesting direction is to consider settings where the actor is aware of the recognizer's presence. Specifically, our approach can be adopted to "transparent planning" (MacNally et al. 2018), where actors choose behaviors that facilitate recognition. These models rely on partially informed agents to be able to choose a behavior that maximizes the information conveyed about their intentions. In such settings, GRD can be viewed as a complementary approach, that can be applied to alleviate the need to completely rely

on the actor, and reduce the number of non-distinctive behaviors. Another variation worth exploring is an interactive setting, where the recognizer can decide which information to reveal based on the actor's actual progress. This would be especially relevant to many realistic settings where the recognizer cannot be assumed to have perfect information about the solver used by the actor. Finally, while we focus on pruning as a way to increase efficiency, other options are possible. In particular, heuristics can be used to estimate the value of a modification, and lead the search in promising directions.

## Acknowledgements

The authors thank Miquel Ramirez, Nir Lipovetzky and Blai Bonet for their helpful comments and suggestions.

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
