# OpenReview forum: "Information Shaping for Enhanced Goal Recognition of Partially-Informed Agents"
_icaps-conference.org/ICAPS/2019/Workshop/HSDIP_

### Official Review · AnonReviewer1 · 2019-04-05
**I recommend acceptance (see below for the brief summary, as it does not fit in this window)**

**Rating:** 7
**Confidence:** 4

**Review:**

Short summary of the paper:
This work lifts the goal recognition design framework to planning domains with incomplete information. The key idea is that an acting agent, the 'actor', plans for a given goal. A second agent, the 'recognizer', who does not know the exact goal of the actor, is allowed to modify the domain through means of sensor extensions: modifications to the observations the actor receives. The goal of the recognizer is to choose sensor extensions such that the goal of the actor becomes clear as early as possible. The paper formally defines the problem, shows that the set of sensor extensions can be retrieved through means of search and shows how to filter relevant sensor extensions by reasoning over the causal graph of the underlying problem. A brief evaluation studies how the presented solution performs over different domains.

Brief summary of my review:
The paper is fluently written and lifts a known problem, goal recognition design, to the incomplete information level. The paper is motivated through simple examples, cites relevant work and formally introduces the underlying problem. A search-based solution method for the special case where the actor relies on the k-planner approach is presented and the paper presents a formally sound pruning procedure to make this search more efficient. While there are minor and less minor points at issue, the overall approach is sound and accompanied by an empirical evaluation which shows that it (sometimes) works. Finally, the paper fits in the scope of the workshop, as it presents a challenging domain for existing combinations of heuristics and search algorithms. Therefore I recommend acceptance.

Quality: The paper is overall sound, for the most part well-written, cites relevant literature and provides a formal and empirical evaluation.
Clarity: Although the structure can be improved and some details are not completely clear in its whole the paper is easy to follow.
Originality: The paper takes a known problem and lifts it to the level of incomplete information. While a bit incremental it is an original approach.

Detailed review:

Introduction / Related Work:
The introduction does a good job in presenting the underlying problem and bringing it in context with related work. I have however an issue with the
motivating example of the hot oven. In my opinion this does not really correlate with the objective of minimising the worst case distinctiveness, as there is no relation to recognizing the goal of the actor early, and preventing him from moving too close to the oven. Overall the paper could do a better job of motivating the reason behind minimising the wcd, and also why we require a domain-independent approach here, if we have to modify the PDDL description anyway.

k-planner Translation / GRD-APK:
The paper would greatly benefit from restructuring the sections such that the k-planner translation is introduced after the definition of the GRD-APK problem (Def. 7). As it stands, when the k-planner translation is introduced is not clear for what it is required. Only later on it becomes clear that the
underlying assumption is that the solver is based on this translation. Additionally, the paper should make this assumption more explicit, as it is
important for the correctness of the proposed solution method later on. Positioning the k-planner translation after Def. 7 allows the paper to clearly
say that in the following the k-planner is assumed to be the underlying solver.

That the recognizer does not see the actor's actions comes a bit sudden and was not really specified before. It is also never mentioned later on. Is this an important point of the model? If so, why is not discussed in more detail?

On the definition of the worst-case distinctiveness:
Initially I assumed that the underlying assumption of the paper is that the acting agent plans optimally, and the worst-case distinctiveness is defined only over histories which are part of optimal policies, which in my opinion is also somewhat suggested by informal parts of the paper (e.g. the introduction of the K_prudent transformation is motivated by reasoning over a cost-minimizing solution, and in Example 1: 'we assume that among cost-minimal plans, it prefers one that requires making as few assumptions as possible').
But part of the evaluation considers a satisficing acting agent, and after a second look at the definition of the worst case distinctiveness there is
indeed no assumption about the set of non-distinctive histories to consider. I think not limiting the worst-case distinctiveness of a problem to optimal
policies makes this a weak measure, because in every type of domain where a policy is not distinctive after a single step and which allows backtracking the wcd will be infinite. The paper gives a very good example by itself: if we do not assume that the actor acts optimally in the Wumpus domain, there is no guarantee that moving right or north reveals the agents goal, as it could just be an action part of a non-optimal plan.

Empirical Evaluation:
The empirical evaluation could more specifically discuss the usefulness of the approach. The wcd reduction on the domains considering an optimal acting agent seems really low, how does it relate to the wcd before modifying the problem? Given my above comment about wcd for satisficing planners, I do not understand how the wcd in these domains is computed. I would assume that the wcd is based on optimal policies, but the histories used to compute the wcd are sub-optimal.
Furthermore, the k-prudent translation was introduced as means to void a solution which favors increasing the cost to the goal. The evaluation should
evaluate and discuss if the k-prudent translation fulfills its intended purpose.

Medium to minor comments:

Introduction:
- Example 1: it would help to explicitly state the wcd for this example
- It took me a second to understand why the plan of the agent moves back from cell (1,3), but I think this setup (of the wumpus and the pit) does not really contribute to the example, as the fact that the agent has to move back is not really required.
- You should not start a sentence with an abbreviation: "wcd is the longest..."
- 'longest number of actions' => a number is not long
- 'It decuces' => deduces
- 'and wcd is minimized' => 'and the wcd is minimized'

Related work:
- 'accounts for setting' => settings
- in-text citations should not have parentheses
- 'one time and offline intervention stage' => I think this could be made more
explicit in the introduction and also later on

Background:
- 'includes two main components;' replace semicolon by a colon
- 'helps the recognizer detect its objective' => to detect
- It would help how the wcd is usually computed
- In the second paragraph of the subsection on contingent planning the order of the definitions seems a bit arbitrary. For example, 'A formula is invariant, a fluent is hidden, an action is applicable' do not really fit together. The applicability should be after the definition of a belief state (then the order of the definitions following a belief state follow the order of Def. 1, i.e. fluents, actions, initial state, goal states).
- 'applying the actions a to each state s in b' => the action

K-planner translation:
- ' cannot visit any state more than once' any state or any belief state?
- re-planing => re-planning (happens a second time later on)
- each sensor (C,L) => each observation
- for variables => for fluents
- invariant clause and ramification actions could be made more clear
- body of a conditional effect is not defined
- definition of hidden: true value unknown for all possible belief states?
- 'belief reachable form b' => belief reachable from b
- 'for every L': what is L? The notation (L) suggests that it is only the second part of the observation pair over all observations, but that is not clear. From
the notion monotonic I would rather assume that a monotonic model implies that whenever a fluent F is known in belief state b there does not exist a belief state b' reachable from b where F is unknown.
- 'over a simple problem P' => this makes formally no sense
- smaller then => smaller than

Evaluating a GR-APK model:
- When reading this part the first time I had the following question:
    - What does generated by an agent mean?
    - 'It satisfies a goal G, if pi is a possible policy to G'. Does 'It' really refer to h? I do not think so, because pi and h are not related. Or do you mean
      'It additionally satisfies', i.e. if it satisfies policy pi it also satisfies G if pi is a policy to G. Also, weak policy? strong? strong cyclic?
- Reading the formal part both becomes clear, but maybe end the sentence with a
colon (or just leave it out, as you define this formally anyway)
- I would rather say h is consistent with pi instead of satisfies

Information shaping:
- Pi(G) is only defined implicitly in the text
- 'to a senor model' => sensor model
- The definition of a sensor extension makes one wonder when a modification is not a sensor extension (formally when it modifies only some sensors but not all), and then why it is important that there exists modifications which are not sensor extensions. If you only talk about sensor extensions and never about other modifications why not only introduce sensor extensions as being an extension of all sensor models?
- b in Def 7. is bold, but afterwards normal font.
- 'from a partial observable to fully observable planning for each goal' => This
is no sentence
- 'is a needed to reach' => is needed
- 'pick is up' => pick it up

- Lemma 1:
The definition of the sensor model only states that it is a set of observations, therefore this definition does not guarantee anything. I think what you instead require is that the underlying POP-det model is simple (as defined by Bonet and Geffner), which guarantees that the model is monotonic.

Design with CG-pruning:
- Why do pruning conditions that preserve optimality not hold in this setting?
- I do not think Example 2 contributes to the understanding of this section. It fits better as an example for the k-planner translation.
- The part which introduces the causal graph is hard to understand for readers not familiar with the CG beforehand. Where do the multi-valued variables come from (also, here the paper talks about variables, whereas before it only talks about fluents)?
- Why are there causal graphs for different iterations if we only require the CG for iteration 0?
- A short discussion how the CG changes over the number of iterations would be interesting
- 'we use the K(P), to transform...' => we use the K(P) transformation, to transform

- The proof sketch of Lemma 3 has multiple grammar and spelling errors
- The proof sketch of Theorem 1 contains the definition of when a pruning function is safe (a proof should not contain a definition)

Evaluation:
- For the representation of the tables used for the evaluation I suggest to take a look into the booktabs package (see also http://cs.brown.edu/about/system/managed/latex/doc/booktabs.pdf)

Literature:
- Ronen Brafman is cited as Ronen Brafman and Ronen I. Brafman
- Some literature includes page numbers some does not (I do not mind either, but please stay consistent)

---

### Official Review · AnonReviewer2 · 2019-04-05
**Initial Findings on Goal Recognition Design with Partially-Informed Agents**

**Rating:** 7
**Confidence:** 3

**Review:**

This paper deals with the goal recognition design problem. It extends
previously used definitions to consider a partially-informed actor whose goal
should be recognized. This is modeled via having a second agent, the
recognizer, who has the (one-time) opportunity to implement so-called
information shaping modifications with the aim of forcing the actor to reveal
her goal earlier. The paper provides the formal framework of the extended GRD
setting (GRD-APK) and presents a simple BFS approach that uses the k-planner
transformation into classical planning problems for solving GRD-APK problems.
The BFS can be enhanced through causal graph-pruning, which is shown to be
safe. The initial evaluation shows that pruning reduces the search effort.

This will be a rather light-weight review for a) I am not very familiar with
existing work on goal recognition design and b) this is a non-archival
workshop.

The problem of GRD has been intensively studied in recent years and is an
interesting application of planning. While the bulk of this paper deals with
the problem formulation, it also discusses aspects of solving GRD-APK
problems through using search and pruning techniques, and hence this paper fits
the scope of the workshop. I am thus voting to accept it.

Generally, my impression at several places was that this paper is in a somewhat
preliminary form. Many of the concepts are described informally and all details
were not always clear for me.

For example, the introduction is, as it should be, high-level, but it is very
long, due to Example 1. After reading Example 1, I understood the setting, but
I didn't understand what "preferring making as few assumptions as possibles"
means concretely and why an actor aiming for G_2 would prefer moving up due to
the added cost of going through the safe location (3, 1), is there some kind of
preference ranking involved?. This was only much later clarified to be simple
tie-breaking of preferring optimal solutions, when explaining the k-planner
translation. Also the discussion of related work was too short for me, the
non-expert, to situate this work and its contributions with respect to previous
work because the section on related work rather reads like a list of references
than an explanation of what was done there ("extended to a variety of GRD
settings").

In the background section, the part on GRD is very short and informal. The
formal part on contingent planning is detailed, well-written and useful, but it
left open (the small question of) how actions are formalized: what are
conditional effects and how are they applied to a state to generate a
successor?

In the GR-APK model (Def. 2), I missed a goal for the recognizer. Shouldn't
this be part of the model? Also using \epsilon for the environment clashes with
\epsilon being the small cost for sensing actions.

Why is the difference between observing actions or transitions so crucial for
the recognizer? This is mentioned three times, I think.

Example 2: did I get this right that K1 can be used only for G1, and hence that
K3 and K4 are useless? What are the exact actions available in that domain?
This would be important also to understand the example causal graph, where I
don't understand why there is no variable for Key1At, assuming that the figure
shows some "subset-CG" relevant to only Key1, and why there are edges from
k_at_(x,y) (the location of the actor, right?) to k_stench_at (or is at(x,y)
use for agent x at location y?) and from k_holding_k_1 to k_at_(x,y).

In the experiments, I wasn't sure what "\delta-wcd" computes: the average
reduction between what? What are "both approaches"? Are they "No pruning" and
"CGPruning" or are they FF and FD? Generally, how can there be positive
\delta-wcd values in comparisons in both directions?

Generally, I have the feeling that the text would benefit a lot from being
proof-read for consistency, small typos, and generally, accurateness of the
description. I'll give a few detailed (minor) comments in the remainder.

- The listing starting with "These include assistive cognition.." is hard to
read (commas with different meanings)

- no comma after "all these settings"

- "accounts for setting where": missing s

- (C, L) vs \langle C, L\rangle

- form -> from

- subsection "Goal Recognition": why is there no space before it? Negative
vspace?

- Def3: h should be specified, otherwise neither n or a nor b is specified.

- We mark -> We write

- either use a comma after "i.e." or use a backslash to avoid the long
whitespace (latex interprets this as a full stop of a sentence). Actually, I
think AAAI Press requires to use \frenchspacing since a while which should
eliminate this problem.

- "which pruning is safe": please define earlier and not only in the proof of
Theorem 1

- Figure 2(left)/(right): use a and b (package subcaption for subfigure + using
and referencing a label should do this)

- space between "(v, v')for"

- observable transformation -> observable problem

- add comma between "execution iteration"

- add comma after "pruning technique, cg-pruning"

- "can be reached via design": can this be explained a bit?

- show that K(P) transformation: missing "the"

- for simple problem: missing s

- can acquired: missing "be"

- Please move all three tables on the last page. I think to get these tables to
their current positions, there has to be done quite some work (like using "!"
or "h!" for figures), and I don't see a reason for having the tables on the
second-to-last page and all of their discussion on the last one.

- no comma in "our dataset, and empirical"

- "we use a PDDL file": what is meant with this?

- "the optimal FD classical planner": does this refer to blind search of FD? Or
some other config?

- "our initial results": why is this initial; what would be changed if this was
a full conference paper?

- missing space between "distinctiveness(wcd)"

- please note that the paper is too long (allowed are 9 pages including
references)

---

### Author Response · Authors · 2019-04-09
**We thank the reviewers for their thoughtful and helpful reviews**


We thank the reviewers for their helpful comments and detailed suggestions. All comments will be addressed in the final version (respecting the space limits). Specific responses to each reviewer are given below.


Reviewer 1:
--------------

Following the reviewer's suggestions, we will apply the following changes.

1. The setup and assumptions we make about the actor's reasoning will be made clearer in the introduction.
2. The contributions of the suggested framework w.r.t previous work on GRD that supported only perfectly informed agents will be emphasized.
3. The actor's planning model will be elaborated to include a detailed description of the actions.
4. Example 2 and its diagram will be clarified to stress the ability to prune redundant sensor extensions (e.g., those that relate to key_3 which doesn't appear in the causal graph of any goal).

GR-APK: In a goal recognition setting (definition 2), the recognizer uses the incoming stream of observations to deduce the actor's goal as soon as possible. Design is used to minimize the worst case distinctiveness, i.e., minimize the maximal number of actions(or path cost) an agent can take before recognition occurs (which is the point in which the recognizer can map the observed sequence to a single goal). This will be clarified.

Observer model: As the reviewer points out, in the deterministic settings with perfect knowledge for the recognizer, there is no real distinction between a sensor model that can monitor the actor's actions (i.e., what the actor decides to do) and state transitions (i.e., what actually happened to the actor). This distinction only becomes relevant in the extensions we intend to suggest for this setting (which are probabilistic or partially observable setting). We see why this comment may be confusing given the current setup and will fix this in the final version.

Evaluation: We will apply all the suggested changes. Specifically, we will clarify that the main objectives of evaluation are to 1) measure \delta-wcd as the reduction in wcd achieved via design, and 2) the savings achieved via our suggested pruning (comparing “No pruning" to
"CGPruning").


We thank the reviewer for all the highlighted typos and minor comments, which will all be addressed.


Reviewer 2:
---------------

1. We will use the comments and reviewers suggestions to clarify the motivation for this work. Specifically, we will clarify that the use of a domain-independent approach allows developing a unified search strategy and allows for the automatic pruning procedure.

2. The suggested structure of the model presentation will be adopted.

3. Sensor model : See response for Reviewer 1.

4. WCD definition: the assumption is that the recognizer knows the actor's solver (sub-optimal or optimal). The wcd takes into account the set of legal behavior of the actor w.r.t to the specified solver.

5. Evaluation: we will specify the ratio of wcd reduction (instead of just the actual reduction, which is what we mention now). Also, as stated above, the role of a sub-optimal solver will be specified, as well as it implications to the running example and results.

All minor comments will be accounted for. Specifically,

6. Sensor extensions: indeed, as stated by the reviewer, our focus here is on information shaping in the form of sensor extensions. However, other changes could be relevant in the suggested setting, depending on the specific application. This would affect the search safety of the pruning techniques we suggest, but not the problem formulation. We will clarify this in the text.

7. Lemma 1 will be corrected to account for the Reviewer’s comment

8. The definition and role of the causal graphs in the discussed context will be clarified.

9. Previous approaches to pruning  in GRD relied on the modifications being monotonic - no modification could make recognition harder, which is not the case here (as we demonstrate in Example 1).

10. We will define safe pruning before the theorem

All the other comments and suggestions will be addressed in the final version.

---

### Meta-Review · Program_Chairs · 2019-04-25

**Recommendation:** Accept
**Confidence:** 5

**Metareview:**

Dear Authors,
thank you very much for your submission. We are happy to inform you that
we have decided to accept it and we look forward to your talk in the workshop.
Please, go over the feedback in the reviews and correct or update your papers
in time for the camera ready date (May 24).
Best regards
HSDIP organizers